# DISCO-10M: A Large-Scale Music Dataset

**Luca A. Lanzendörfer**
ETH Zurich
Zurich, Switzerland
`lanzendoerfer@ethz.ch`

**Florian Grötschla**
ETH Zurich
Zurich, Switzerland
`fgroetschla@ethz.ch`

**Emil Funke**
ETH Zurich
Zurich, Switzerland
`efunke@ethz.ch`

**Roger Wattenhofer**
ETH Zurich
Zurich, Switzerland
`wattenhofer@ethz.ch`

## Abstract

Music datasets play a crucial role in advancing research in machine learning for music. However, existing music datasets suffer from limited size, accessibility, and lack of audio resources. To address these shortcomings, we present DISCO-10M, a novel and extensive music dataset that surpasses the largest previously available music dataset by an order of magnitude. To ensure high-quality data, we implement a multi-stage filtering process. This process incorporates similarities based on textual descriptions and audio embeddings. Moreover, we provide precomputed CLAP embeddings alongside DISCO-10M, facilitating direct application on various downstream tasks. These embeddings enable efficient exploration of machine learning applications on the provided data. With DISCO-10M, we aim to democratize and facilitate new research to help advance the development of novel machine learning models for music.[1]

## 1 Introduction

Music is a universal language that has captivated and inspired humans for millennia. With the rapid advancements in technology, the domain of music has become increasingly interconnected with the field of machine learning, opening up new possibilities for music analysis, music recommendation systems, and the generation of novel compositions. Central to these developments are large and high-quality music datasets that serve as the foundation for training and evaluating modern machine learning models.

The recent breakthroughs achieved by large language models in the textual domain and diffusion models in the visual domain have been largely attributed to the availability of massive datasets. These datasets have played a pivotal role, enabling large models to learn intricate patterns and generate coherent output. However, in the realm of music, the availability of large-scale datasets has remained relatively limited, impeding the same advancements of research in this domain.

In the field of text processing, datasets such as *CommonCrawl* [1] and *the Pile* [13] have made substantial amounts of written content available for training large language models. For instance, *the Pile* contains an extensive dataset of 825 gigabytes of text, while *CommonCrawl* contains a staggering 3.3 terabytes of text. These datasets capture a significant portion of the writing available on the Internet, providing an abundant resource for training and evaluating language models.

---

[1] `https://huggingface.co/DISCOX`

Table 1: Comparison of public music datasets.

| Dataset | # Clips | # Artists | Track duration | Year | Audio |
|---|---|---|---|---|---|
| GTZAN | 1,000 | 300 | 30s | 2002 | yes |
| MagnaTagATune | 25,863 | 230 | 29s | 2009 | yes |
| MTG-Jamendo | 55,609 | 3,565 | full-length | 2019 | yes |
| Free Music Archive | 106,574 | 16,341 | full-length | 2017 | yes |
| Music4All | 109,269 | 16,269 | 30s | 2020 | yes |
| Million Song Dataset | 1,000,000 | 44,745 | full-length | 2011 | no[2] |
| AudioSet[1] | 1,011,305 | - | 10s | 2017 | no[2] |
| AcousticBrainz | 2,524,739 | - | - | 2015 | no |
| DISCO-10M | 15,296,232 | 400,047 | full-length | 2023 | no[2] |

[1] Only 1,011,305 out of 2,084,320 clips are labeled as `Music`.
[2] Audio not directly available, can be downloaded from YouTube.

Similarly, in the visual domain, there exist various large datasets, such as the famous ImageNet [9] dataset, Open Images [19], Laion-400M [26], and Laion-5B [27] dataset, which all contain massive amounts of images. These datasets have been instrumental in the advancements of computer vision research. Especially Laion-5B, with its five billion image-text caption pairs, has been essential for the success of current text-to-image generative models.

Contrary to the textual and visual domains, existing music datasets face several limitations that hinder the progress of research in machine learning for music. One of the primary challenges is the limited size of available datasets, which restricts the diversity and representativeness of the musical content that can be analyzed, as well as the use-cases that can be tackled. Additionally, accessibility to these datasets has been a concern, with many of the state-of-the-art models for audio and music being trained on proprietary datasets that are not accessible to the broader research community [11, 16, 37, 8, 18, 2, 17]. Moreover, the scarcity of available audio recordings poses a significant hurdle in obtaining datasets and, therefore, training novel machine learning models for music.

To address these shortcomings, we introduce DISCO-10M, a novel and extensive music dataset that surpasses the largest previously available music dataset by an order of magnitude. By curating DISCO-10M, our objective is to overcome the limitations of existing datasets and provide researchers with a rich and diverse collection of music data, enabling further advancements in the field of machine learning for music.

In addition to the extensive music dataset, we also provide precomputed audio embeddings alongside DISCO-10M, which we obtain from a pre-trained open-source CLAP model [35]. By including precomputed audio embeddings, we facilitate the direct application of machine learning models on various downstream tasks, eliminating the need for time-consuming audio retrieval and embedding computation, and reducing the barrier to entry for researchers in the field.

The availability of DISCO-10M aims to democratize access to large and high-quality music data for the research community. We envision that this dataset will serve as a catalyst for new research endeavors, inspiring researchers to explore novel machine learning models, techniques, and applications in the domain of music. By advancing our understanding of music through machine learning, we can foster the development of innovative music analysis tools, personalized recommendation systems, and creative music generation models.

## 2   Related Work

**Music Datasets.** We list some of the larger music datasets which have influenced our work. While GTZAN is small compared to other music datasets, it was one of the first datasets, originally intended for music genre recognition [32]. The dataset contains 1000 files, each with a 30-second music clip. Every song belongs to one of ten categories: Blues, Classical, Country, Disco, Hip Hop, Jazz, Metal, Pop, Reggae, and Rock. The data was gathered from various sources, including personal CDs, radio, and microphone recordings, and therefore varies greatly in quality. Furthermore, GTZAN does not

provide other metadata, such as artist or album names. Even though GTZAN is small compared to more recent datasets, it remains a widely used dataset for music information retrieval tasks [31].

MagnaTagATune is a dataset that contains 25,863 music clips, each 29 seconds long [20]. The music clips are extracted from 5223 unique songs across 230 artists. The dataset provides 188 tags for each music clip, crowd-sourced from TagATune.[2] TagATune is an online two-player game where each player creates tags for a music clip. Afterward, players must decide based on both their created music tags if they were presented with the same music clip. If a music clip receives the same tag from both players, the tag is added to the music clip in the dataset.

Million Song Dataset (MSD) contains metadata and audio features for one million songs but does not contain the audio files [4]. The dataset was initially created to bridge the gap between academia and industry by enabling academia to benchmark music information retrieval algorithms on industry-scale datasets.

AcousticBrainz is an open source and community-driven database of 2.5 million music entries [22]. While it is the largest music database, it does not distribute music or store links to third-party sites where the music can be found. Like MSD, AcousticBrainz stores audio features and metadata such as overall loudness, rhythm, genres, and instrumentation.

Free Music Archive (FMA) dataset was the first publicly available dataset to contain more than 100,000 freely available full-length music files across 16,000 artists [7]. The dataset is sourced from a freeform radio station of the same name,[3] which provides music released under the Creative Commons license. FMA provides fine-grained hierarchical genre metadata across 161 genres and contains high-quality audio.

AudioSet is a dataset containing two million YouTube video IDs with associated audio classes [14]. AudioSet was created as a dataset for audio event detection and to train and evaluate machine perception. Each dataset entry has a YouTube ID and a start and end timestamp. Furthermore, the dataset contains an audio label and a short audio description. However, since we are specifically interested in music datasets, it should be noted that only one million IDs contain the music label. Similar to MSD, the audio needs to be downloaded from YouTube as it is not provided with the dataset.

MTG-Jamendo was introduced as a dataset for automatic music tagging [5]. The data is a subset of music sourced from the Jamendo website,[4] which contains royalty-free music. Jamendo offers music for commercial use, which means the music is generally of higher quality. The dataset contains music tracks with 692 tag annotations, consisting of genre, instrument, and mood tags.

Music4All is a dataset that contains 109,269 music clips across 16,269 artists [21]. Each music clip is 30 seconds long and is cut from the middle of the original track. The dataset was created by collecting 15,602 users and their listening histories from last.fm.[5] The music from the listening histories was downloaded from YouTube and post-processed by adding various features. The dataset contains 26 different features for each sample, such as the music clips and tags from last.fm, audio features from Spotify, and lyrics from MusiXMatch.[6]

**Embedding models.** Learning latent representations of data has been an active field of research over the past decade [3, 23, 15, 33, 6]. One of the more recent advances in representation learning comes from language-image pre-training, where a CLIP (Contrastive Language–Image Pre-training) model was trained on 400M pairs of language-image samples and obtained state-of-the-art zero-shot classification results [24]. The resulting model learned a joint embedding space for both images and texts. This idea was extended to audio with CLAP (Contrastive Language–Audio Pre-training) [12]. We use Laion-CLAP [35], an open-source alternative to CLAP. Although we could also embed text with CLAP, we decided to use it only for audio. For text, we use Sentence BERT [25]. Sentence BERT is trained by fine-tuning a BERT model via a siamese and triplet network to obtain semantically meaningful text embeddings [34, 10].

---

[2]https://tagatune.org/

[3]https://freemusicarchive.org/

[4]https://www.jamendo.com/

[5]https://www.last.fm/

[6]https://www.musixmatch.com/

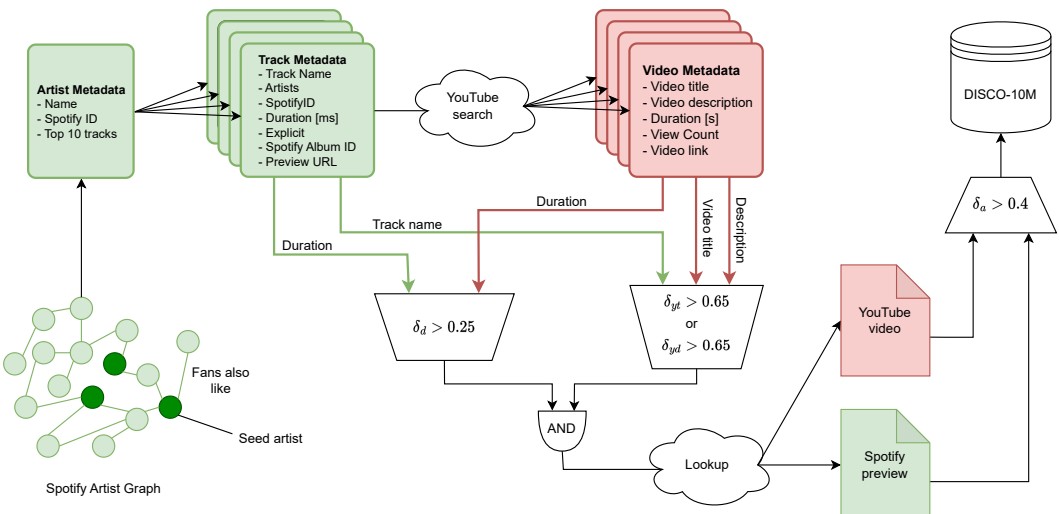

Figure 1: Overview of pipeline to create DISCO-10M. Starting from a seed list of artists, we crawl the Spotify artist graph and collect related artists and their metadata. For each artist, we collect the metadata of up to ten "top tracks" and use them to search for up to twenty YouTube videos. The YouTube metadata, together with the Spotify metadata, are used to filter out datapoints according to their duration, title, and description similarity. The remaining matches are downloaded from Spotify and YouTube and compared. A datapoint is discarded if the similarity is below the threshold.

**Music Providers.** We use Spotify, one of the most popular music, podcast, and audiobook streaming platforms, with over 515 million users, 9 million artists, and more than 100 million available songs [28, 30]. Furthermore, we use YouTube, one of the largest on-demand video streaming platforms, with more than two billion monthly users and more than 500 hours of video uploaded every minute [36]. YouTube offers a wide variety of content on its platform and allows anyone to upload their artistic creations to YouTube. This allows us to find audio samples for relatively unknown artists since the barrier to entry on YouTube is considerably lower than on Spotify. We use YouTube to expand our Spotify data with music videos and additional metadata.

## 3 Data Collection

Figure 1 gives an overview of the data collection process for DISCO-10M. We compute a list of Spotify artists generated with a breadth-first search over related artists, starting from a seed list of artists that covers multiple genres. We then take the most popular songs for every artist, according to Spotify, and search for the songs on YouTube. At this point, we end up with a one-to-many mapping of songs to YouTube search results that we filter to improve the quality of the matches. In the first filtering step, we apply filters based on duration, the similarity between Spotify song title and YouTube title, and the similarity between the song title and the video description. We then download Spotify previews and YouTube videos for these prefiltered datapoints to generate CLAP audio embeddings. The similarity of these embeddings is used in the last filtering step. A full list of all metadata columns that we provide for DISCO-10M can be found in Appendix B.

### 3.1 Spotify Artist Graph

Our goal is to obtain songs from diverse artists that span multiple genres. To decide which artists to consider, we start with a hand-curated list of seed artists (cf. Appendix A). We chose popular artists that are influential and represent their genre. To explore additional artists, we look for artists that "fans also like", a feature provided by Spotify [29]. Starting from the seed list, we explore these related artists one hop at a time, adding all related artists to the set of artists we already know. In other words, we obtain a directed graph if we represent artists as nodes and add edges between them according to their *related artist* relationship. We now consider the nodes representing artists in our seed list and run a breadth-first search on this graph, discovering a subset of all Spotify artists.

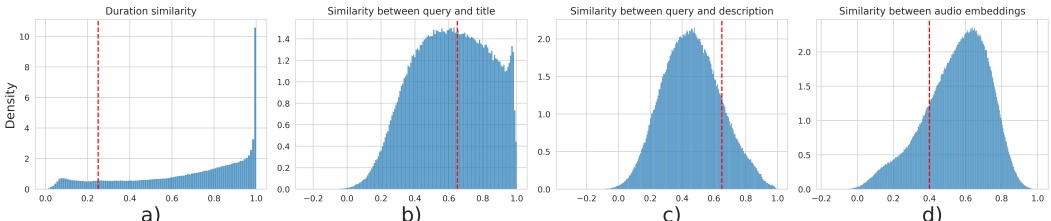

Figure 2: Distribution of data before each filtering step. We first filter the data according to duration similarity a). The data is then filtered if it is either below the title similarity threshold b) or below the description similarity threshold c). Finally, the data is filtered according to the audio similarity d).

Table 2: Examples for different levels of text similarity between search query and YouTube video title

| Search query | YouTube video title | Text similarity |
|---|---|---|
| Take On Me Superton | Danelectro Doublenack Guitar&Bass | 0.382 |
| Lead Me Eddie Neblett | Eddie Neblett - Reset (Official Lyric Video) | 0.663 |
| Che pyhare mombyry - Version polac Perfil | Che pyhare mombyry (Version polac) | 0.964 |

Table 3: Examples for different levels of text similarity between search query and YouTube video description

| Search query | YouTube Description Snippet | Text similarity |
|---|---|---|
| Higher Ally Brooke | Listen with headphones on!! — CLICK!! — Hello everyone! My name is Dewi, and I'm making Youtube Empty Arena, Slowed... | 0.115 |
| I Feel You Toly Braun | #tolybraun#deephouse. | 0.657 |
| Nobody Knows When You're Down And Out Scrapper Blackwell | Nobody Knows You When You're Down and Out by Scrapper Blackwell. | 0.955 |

We stop after we find about 400,000 unique artists. While we collect this data, we also accumulate metadata provided by Spotify; this includes the artist name, artist music genres, artist popularity on Spotify, and the total number of Spotify followers for each artist in our list.

We take up to 10 "top tracks" per artist to sample songs for our dataset. The "top tracks" classification is provided by Spotify and is based on the listening behavior of users in the US market. We end up with approximately 2.5M unique songs and store metadata such as track name, album ID, artist ID, explicit flag, track duration, track release date, and a track preview URL. After collecting the song information, we can use it to search for videos on YouTube with a lookup containing the name of the song and the name of the artist.

### 3.2 Filtering

After the previous steps, we are left with a one-to-many mapping of Spotify songs to YouTube search results. After cleanup of the data, we keep the following fields: the Spotify track ID, its corresponding track metadata, artist metadata from Spotify, and a YouTube search result with its corresponding metadata.

When keeping all search results from YouTube, we have 46.8M matches. To improve the quality of the dataset and remove bad matches, we filter the dataset by duration similarity $\delta_d$, followed by text similarity $\delta_{yt}$ and $\delta_{yd}$, and finally by audio similarity $\delta_a$. We empirically find that the following filtering thresholds work well:

$$(\delta_d > 0.25) \wedge (\delta_{yt} > 0.65 \vee \delta_{yd} > 0.65) \wedge (\delta_a > 0.4)$$

Applying duration and text filtering stages removes approximately half of all entries from our dataset, which results in the dataset containing 20M entries before audio similarity filtering. We filter the remaining entries based on their similarity between the Spotify preview and the YouTube audio embedding.

We demonstrate the effect of our filtering pipeline on the data distribution in Figure 2. In a), we observe a small peak in the duration similarity at around 0.05. This is because YouTube search results contain relatively long videos compared to Spotify song duration. These long videos tend to be multi-hour versions of songs, full movies, or news broadcast live-streams. Therefore, we set the filtering cut-off in terms of duration such that YouTube videos are at most four times as long as the Spotify song.

The density distributions shown in Figure 2 b), c) are the basis for the title and description filtering. We found relatively strict filtering to work best as it only keeps good matches and reduces the number of datapoints we have to process.

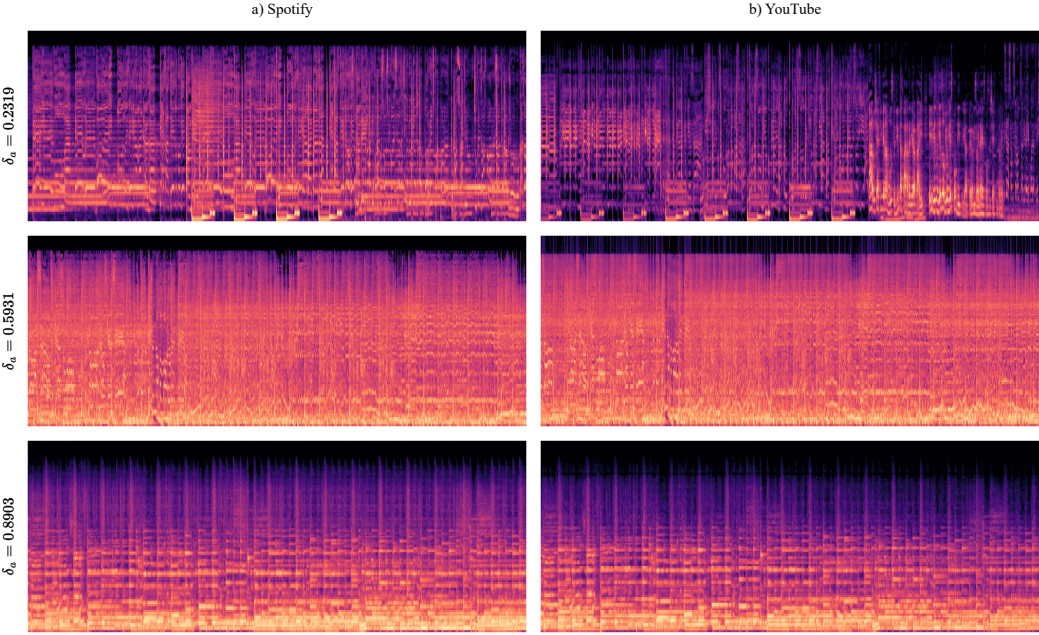

Figure 3: Comparison of audio similarity between Spotify preview audio and YouTube audio. $\delta_a$ denotes the cosine similarity of the audio embedding. We use Log-Mel Spectrograms to visualize audio. We empirically observe that the similarity of our audio embeddings is related to the similarity of the Log-Mel Spectrograms and that the similarity increases when the spectrograms are closer to each other.

Using the OR operator in the comparison, we still allow some titles and descriptions with lower similarities to pass to the next filtering stage, as long as one of the two similarities is high enough. Table 2 provides examples of search queries and corresponding YouTube video titles, where matches with similarity scores above our threshold will continue to the next stage. The same is shown in Table 3 for YouTube video descriptions.

**Duration Similarity**   The duration similarity $\delta_d$ is computed from the Spotify track duration $t_s$ and the length of the YouTube video $t_y$ as follows:

$$\delta_d = 1 - \frac{|t_s - t_y|}{\max(t_s, t_y)},$$

where $|\cdot|$ denotes the absolute value. We chose this similarity metric because it intuitively captures the difference in audio duration, expressing the similarity as a percentage difference between longer and shorter tracks. This allows for fine-granular filtering of the data, where track $t_s$ and $t_y$ have the same duration when $\delta_d = 1$, and very dissimilar duration as $\delta_d \to 0$.

**Text Embedding Similarity**   We use the YouTube search query, the YouTube video title, and the YouTube video description  to measure the similarity and thus the match of the resulting video compared to our search query. All three texts are embedded into a latent space using Sentence BERT [25]. To measure the similarity of embeddings, we use a cosine similarity between the search query and the video title, denoted as $\delta_{yt}$, and the cosine similarity between the search query and the video description, denoted as $\delta_{yd}$. This allows us to filter poor matches in our data before downloading Spotify previews and YouTube videos, which are used to compute the audio embeddings. We qualitatively compare text embedding matches for YouTube video titles in Table 2 and for YouTube descriptions in Table 3.

**Audio Embedding Similarity**   After applying text-based filters, we download Spotify audio track previews and YouTube videos to compute audio similarity and further filter the dataset. The pipeline to compute the audio embedding similarity for one datapoint is as follows: We download the Spotify

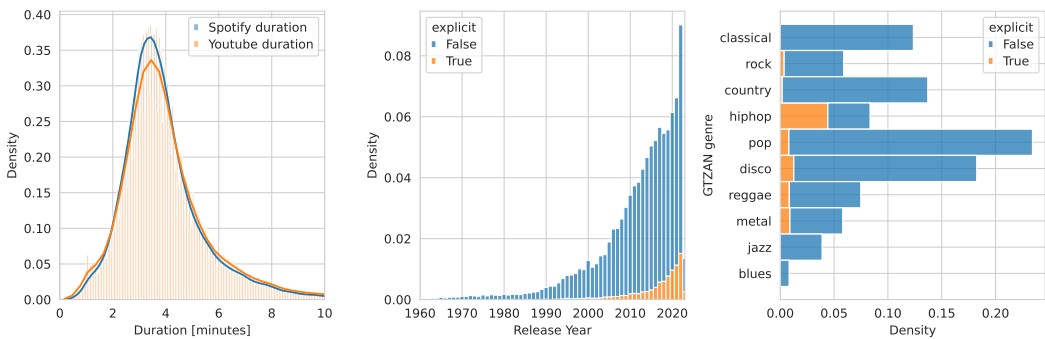

Figure 4: **Left** Song duration for Spotify and YouTube after applying our filtering pipeline. The average track length is 3.5 minutes. We observe a strong correlation between Spotify and Youtube track duration, in part due to the duration similarity filtering. **Middle** Number of tracks and number of explicit tracks released per year. **Right** Genre distribution of GTZAN genres in our dataset. The genre embeddings were computed using a CLAP encoder on the sentence `This audio is a <genre> song.` Each song is mapped to a genre according to the largest cosine similarity between the song embedding and the genre embedding.

preview of the song together with the corresponding YouTube video. Both are then fed into the Laion-CLAP audio encoder model to produce an audio embedding for each audio sample. Once we have the embedding of the Spotify preview audio snippet and the YouTube audio, we compute the cosine similarity of the embedding vectors. We denote the audio similarity as $\delta_a$.

We qualitatively evaluate the audio embedding and similarity scores in Figure 3. To compare the similarity between the two audio tracks, we use the perceptually relevant log-mel spectrogram. Since the Spotify preview is only 30 seconds, and the YouTube audio contains the entire track, we apply a template matching algorithm to find the best overlapping spectrogram section of 30 seconds in the YouTube audio. We observe an increase in spectrogram similarity and, therefore, audio similarity as $\delta_a$ increases.

### 3.3 Subsets

In addition to the DISCO-10M dataset, we propose several subsets with varying sizes and quality presets to enable fast prototyping as well as development with less powerful hardware.

- **DISCO-10K-random** contains 10,000 random samples from DISCO-10M. This split is mainly intended for prototyping and tasks that require substantially less data. It also enables training with fewer computational resources and hardware with memory limitations, even when all audio embeddings are loaded into memory.

- **DISCO-200k-random** contains 200,000 random samples and was used for the data analysis in the following section. The dataset is small enough to remain manageable on common hardware while still being a statistically meaningful representation of our dataset.

- **DISCO-200k-high-quality** contains a selective subset of DISCO-10M with strict filtering based on duration, text, and audio similarity values. The matches of Spotify titles to YouTube videos are better than in DISCO-10M, resulting in a dataset that can be used for tasks where high-quality matches are required.

## 4 Data Analysis

We analyze the distributions of the duration of the song and the release year. Due to our similarity filtering, duration distributions for Spotify and YouTube match each other, with an average song duration of about 3.5 minutes. As the release dates are taken from Spotify metadata, they represent the date of the audio recording. The earliest date of recordings in our dataset is in the 1960s; while there are some individual songs that have an earlier date in the metadata, we remove them as outliers. We also observe that the number of songs released per year grows rapidly. This coincides with the

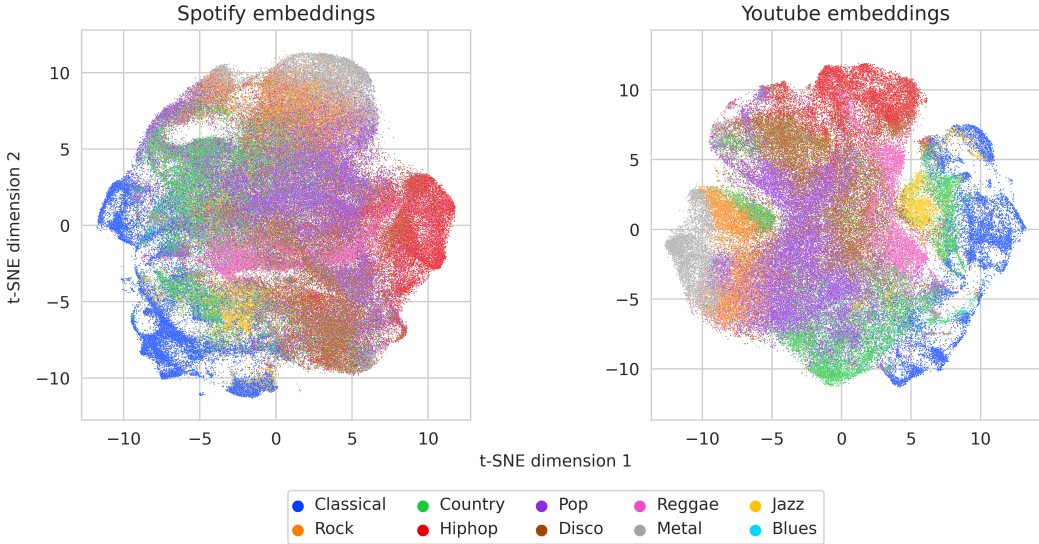

Figure 5: t-SNE plots for Spotify preview embeddings and Youtube audio embeddings computed with CLAP. Colors represent GTZAN genres that were computed from the Spotify embeddings with zero-shot genre classification. The t-SNE plots show the relative positions of music samples, where samples with similar embeddings in the CLAP latent space are located closer to each other and dissimilar points farther apart. We observe that genres are well separated for both the YouTube embeddings and Spotify embeddings. Genres like classical music and metal lie far apart, whereas genres such as metal and rock lie close together.

proliferation of digital music. The cutoff date for our dataset is Spring 2023. We plot song duration and release years for songs in DISCO-10M in Figure 4.

To analyze the genre distribution in DISCO-10M, we utilize the precomputed CLAP embeddings of the 30-second Spotify previews for zero-shot genre classification. We do so by first computing text embeddings of genre prompts and then identifying the closest genre embedding using cosine similarity in the shared latent space for every song. We compute the genre text embeddings with the prompt: `This audio is a <genre> song.` Note that the genre classification is based on YouTube embeddings, of which multiple variations can exist per song, as DISCO-10M contains approximately 2.6M Spotify songs that are mapped to 15.3M music videos on YouTube.

Figure 4 shows the distribution of GTZAN genres on a representative slice of the dataset. Based on this classification, DISCO-10M is mainly composed of pop and disco music, but offers a broad spectrum overall. We further depict the proportion of explicit songs for each genre. We find that Hip-Hop songs have the highest percentage of explicit songs in our dataset, at around 50%.

To evaluate the quality of our genre classification, we compute t-SNE plots on the Spotify and Youtube embeddings, together with their assigned genre (see Figure 5). We observe a separation of genres, where the overlap of neighboring genres is also reasonable, e.g., for metal and rock. Classical music is the most fragmented genre, whereas other genres build more contiguous areas. This aligns well with our expectations regarding the genre distribution, lending it a certain degree of credibility.

## 5 Limitations and Ethical Discussion

We collected data from our server located in Zurich, Switzerland, between January and June 2023. As search results are impacted by time and location, our search results are likely different from other locations, adding bias to our dataset and hindering reproducibility. The artist graph we obtained from Spotify is also subject to changes, and our dataset represents a snapshot of the graph during the mentioned time frame.

Furthermore, the dataset we distribute contains links to online resources for songs and videos that we do not control. These resources can change over time – they may be removed or replaced entirely

by YouTube, Spotify, or their creators. However, since DISCO-10M can contain multiple matching YouTube results for a single song, this redundancy helps counteract dataset degradation over time. Due to this redundancy, we may also include search results that might not represent the original song but something related to it. We believe these samples are still valuable, but users must be aware of this limitation. There is no guarantee for the correctness of the match between the Spotify track and the corresponding YouTube video. For tasks where higher match quality is important, we provide the subset DISCO-200k-high-quality.

The metadata for each song in our dataset is either produced by us (i.e., audio and text embeddings) or available online (i.e., song duration, video duration, view count, etc.). We only provide links to publicly available sources and do not own the copyright of any music referenced in the dataset.

Additionally, the list of artists collected by crawling the "fans also like" section on Spotify might be skewed towards certain genres, artists, or cultures, which can result in a biased representation of music. This can perpetuate stereotypes or hinder the inclusivity of diverse musical expressions.

Another concern stems from restrictions during the data collection process: We only access, download, and process YouTube videos that are not age-restricted. This can lead to some bias in the data, although it should be mentioned that not all Spotify songs that are marked as explicit are also restricted on YouTube. The explicit flags provided by Spotify and the exclusion of age-restricted YouTube videos allow for easier filtering of non-harmful content, but we cannot guarantee that the dataset does not contain hateful or offensive content after filtering. Therefore, in its current form, we strongly advise using this dataset for academic purposes only.

## 6   Conclusion

We introduce DISCO-10M, an extensive and novel music dataset that surpasses existing music datasets by an order of magnitude. The data collection process involves leveraging popular songs from Spotify and identifying corresponding YouTube videos through careful search result filtering, ensuring the inclusion of high-quality data. To enhance the usability of the dataset, we provide embeddings for both song previews from Spotify and embeddings from YouTube videos, enabling researchers to apply their own thresholds for further filtering and facilitating efficient exploration of downstream tasks and data analysis. Our analysis demonstrates DISCO-10M's broad coverage of songs from various genres.

Moreover, the paper highlights the limitations of the dataset and addresses the ethical impact associated with its creation and usage. Recognizing the importance of ethical considerations, we encourage responsible and mindful utilization of DISCO-10M, acknowledging the potential implications and ensuring that future research and applications derived from the dataset are conducted with care.

Overall, DISCO-10M represents a valuable resource for the research community, providing a large-scale music dataset that can be used to advance various aspects of music analysis, personalized recommendation systems, or creative music generation models. Its comprehensive coverage, combined with the availability of embeddings, contributes to the broader field of music research by democratizing access to large-scale music datasets.

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

## A  Seed artists

We use the following artists as a seed list for the traversal of the artist graph on Spotify.

```
starting_artists = [
    '2ye2Wgw4gimLv2eAKyk1NB', # Metallica
    '3WrFJ7ztbogyGnTHbHJFl2', # The Beatles
    '74ASZWbe4lXaubB36ztrGX', # Bob Dylan
    '1Xyo4u8uXC1ZmMpatF05PJ', # The Weeknd
    '4y6J8jwRAwO4dssiSmN91R', # Muddy Waters
    '776Uo845nYHJpNaStv1Ds4', # Jimi Hendrix
    '6kACVPfCOnqzgfEF5ryl0x', # Johnny Cash
    '67ea9eGLXYMsO2eYQRui3w', # The Who
    '7dGJo4pcD2V6oG8kP0tJRR', # Eminem
    '5pKCCKE2ajJHZ9KAiaK11H', # Rihanna
    '0dmPX6ovclgOy8WWJaFEUU', # Kraftwerk
    '1ZwdS5xdxEREPySFridCfh', # 2pac
    '0kbYTNQb4Pb1rPbbaF0pT4', # Miles Davis
    '6tbjWDEIzxoDsBA1FuhfPW', # Madonna
    '7guDJrEfX3qb6FEbdPA5qi', # Stevie Wonder
    '2QsynagSdAqZj3U9HgDzjD', # Bob Marley
    '5aIqB5nVVvmFsvSdExz408', # Johann Sebastian Bach
    '0Kekt6CKSoOm5mivKcoH51', # Sergei Rachmaninoff
]
```

## B  Columns in DISCO-10M Dataset

```
dataset_columns = [
    'video_url_youtube',
    'video_title_youtube',
    'track_name_spotify',
    'video_duration_youtube_sec',
    'preview_url_spotify',
    'video_view_count_youtube',
    'video_thumbnail_url_youtube',
    'search_query_youtube',
    'video_description_youtube',
    'track_id_spotify',
    'album_id_spotify',
    'artist_id_spotify',
    'track_duration_spotify_ms',
    'primary_artist_name_spotify',
    'track_release_date_spotify',
    'explicit_content_spotify',
    'similarity_duration',
    'similarity_query_video_title',
    'similarity_query_description',
    'similarity_audio',
    'audio_embedding_spotify',
    'audio_embedding_youtube',
]
```

## C  Additional Examples: Audio Similarity Comparison

We demonstrate the results of our audio similarity approach on five additional samples (see Figure 6). Similarly to the spectrograms presented in Section 3.2, we also observe an overlap for these samples when the audio similarity is above $\delta_a > 0.4$. Even when two music snippets have the same frequency characteristics, there might still be small differences. This can be explained in part due to the audio

quality of a YouTube video, which is dependent on the quality selected by the person uploading the video and can, therefore, vary greatly, unlike the audio quality of Spotify. This difference can be seen best in the high-frequency content of the spectrogram, which tends to be weaker and less pronounced in the YouTube audio samples. We notice a strong dissimilarity in the first example between the Spotify preview audio spectrogram and the YouTube audio spectrogram. This is reflected by the low similarity score of $\delta_a = 0.1985$.

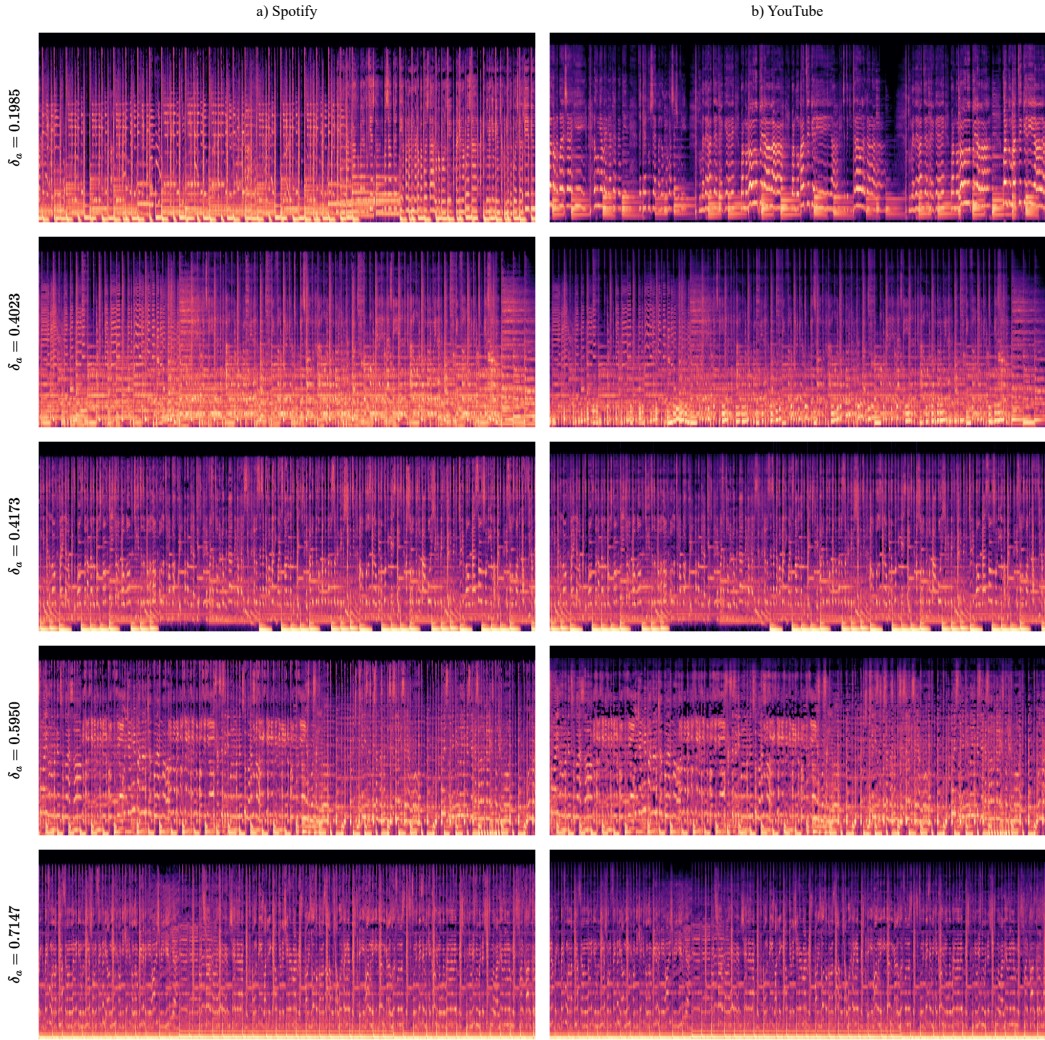

Figure 6: Comparison of audio similarity between Spotify preview audio and YouTube audio. $\delta_a$ denotes the cosine similarity of the audio embedding.We observe that the similarity of our audio embeddings is related to the similarity of the Log-Mel Spectrograms, and that the similarity increases when the spectrograms are closer to each other.

## D   FMA Genre Analysis

We repeat the zero-shot genre classification from Section 4 for the 16 FMA root genres. Figure 7 shows the genre distribution for the FMA genres, while Figure 5 depicts the same t-SNE plot as shown in Figure 5 for these genres. We can observe that our results between overlapping genres are consistent, and although there are more genres, we can observe meaningful relationships between them.

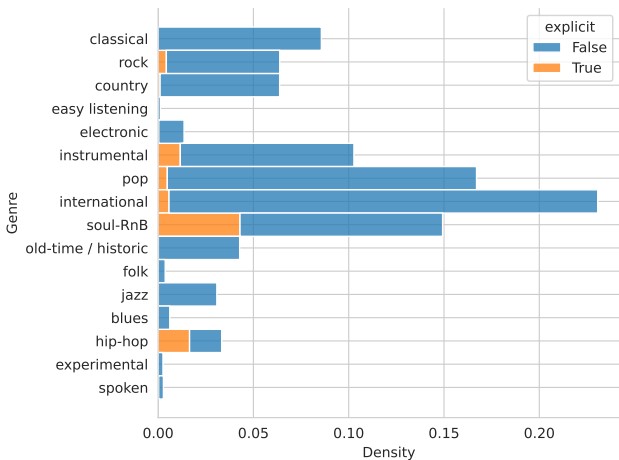

Figure 7: Genre distribution of FMA root genres in our dataset. The genre embeddings were computed using a CLAP encoder on the sentence `This audio is a <genre> song`. Each song is mapped to a genre according to the largest cosine similarity between the song embedding and the genre embedding.

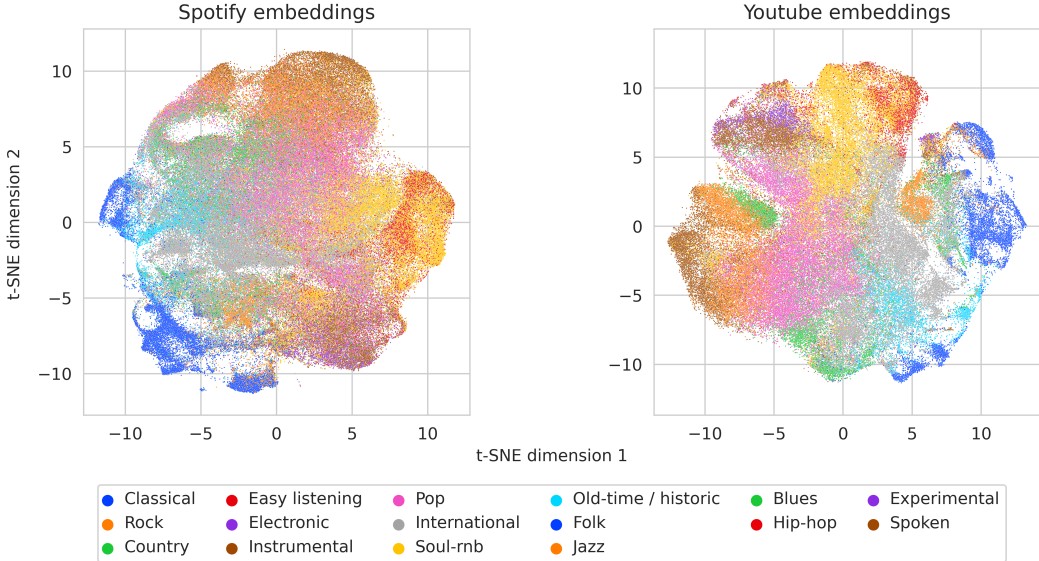

Figure 8: t-SNE plots for Spotify preview embeddings and YouTube audio embeddings computed with CLAP. Colors represent FMA root genres that were computed from the Spotify embeddings with zero-shot genre classification. The t-SNE plots show the relative positions of music samples, where samples with similar embeddings in the CLAP latent space are located closer to each other and dissimilar points farther apart. We observe that genres are well separated for both the YouTube embeddings and Spotify embeddings.

# E    Subset Details

As described in Section 3.3, we provide three different subsets of DISCO-10M.

DISCO-10K-random contains 10,000 random samples from DISCO-10M. We select the samples randomly from unique Spotify track IDs, meaning that the dataset will contain exactly one YouTube video link per Spotify song.

DISCO-200k-high-quality contains 200,000 high-quality samples filtered more strictly to improve the quality of the matches. We created this subset by filtering DISCO-10M with $(\delta_a > 0.7) \wedge (\delta_{yt} > 0.8)$.

# F    Audio Characteristics

The total playtime of YouTube videos in DISCO-10M is approximately 1,062,604 hours or 121 years.

We provide further insights on the audio attributes of the sample rate, MP3 file bitrate, and number of channels based on the DISCO-10K-random subset. When downloading Spotify previews in MP3 format, the audio quality and characteristics remain consistent. All audited samples share common attributes: a sample rate of 44.1 kHz, a bitrate of 96 kbps, and a 2-channel stereo setup.

In the case of audio from YouTube, there is a noticeable resemblance, albeit slightly less uniform. 99.78% of the examined videos employ a standard stereo 2-channel configuration. 0.18% of videos are mono channel, while 0.04% utilize 6-channel surround for audio output. 99.96% of videos have a sample rate of 44.1 kHz, aligning with the settings of Spotify previews—the remaining 0.04% deviate by having a sample rate of 48 kHz.

# I    Ethical Considerations

Copyright and licensing agreements are a complex issue, particularly when it comes to big data collection for training large machine learning models. We acknowledge the concerns of artists regarding the potential negative impact on their artistic work. However, we believe that openly sharing such data helps democratize the research of music understanding and music creation.

To address artists who disagree with our assessment, we offer two options for reconciliation. First, artists can contact us to request the removal of links associated with their art from our dataset.[7] Second, artists may choose to take down their YouTube video or Spotify song from the respective platform, rendering the link contained in DISCO-10M invalid. It is important to note that our guarantee applies solely to our dataset, while other entities who hold private audio datasets may not offer the same level of control.

Creating a dataset of this magnitude is achievable using publicly available tools and a reasonable timeframe. By making our dataset open-source, we also aim to raise awareness on the ease of creating big datasets and uncover the potential existence of similar datasets held by private institutions. Our goal is to provide an opportunity for anyone interested to explore ideas with this dataset, and to enhance our understanding of music creation and safety with large datasets. We emphasize that this dataset serves as a starting point, pushing the boundaries and fostering research of enhanced datasets for various tasks in machine learning for music. Access to such extensive datasets is crucial, not only in the visual domain as demonstrated by Laion-5B, but also in the domain of music.

In summary, our ethical framework emphasizes the importance of respecting artists' concerns, providing options for data exclusion, promoting transparency in dataset creation, and facilitating meaningful exploration of ML-assisted music creation while prioritizing safety considerations.

---

[7]Contact via disco.dataset@ethz.ch email

# J Datasheet for Datasets

## J.1 Motivation

1. **For what purpose was the dataset created?** Was there a specific task in mind? Was there a specific gap that needed to be filled? Please provide a description.

   - We want to provide an open-source large-scale music dataset for the research community. Such large datasets do not yet exist in this domain, and we believe they are needed to democratize innovation in music research and ML-assisted music creation. Working with large data also has inherent risks, which are better analyzed openly by a large community rather than by private institutions behind closed doors.

2. **Who created the dataset (e.g., which team, research group) and on behalf of which entity (e.g., company, institution, organization)?**

   - The authors created the dataset as part of their ongoing research at ETH Zurich.

3. **Who funded the creation of the dataset?** If there is an associated grant, please provide the name of the grantor and the grant name and number.

   - N/A

4. **Any other comments?**

   - No.

## J.2 Composition

1. **What do the instances that comprise the dataset represent (e.g., documents, photos, people, countries)?** Are there multiple types of instances (e.g., movies, users, and ratings; people and interactions between them; nodes and edges)? Please provide a description.

   - We share 15,296,232 YouTube links to music with metadata and associated Spotify music metadata. In addition, we provided similarity measures between the YouTube video title, the YouTube description, the Song title, and the name of the artist. Additionally, contribution includes providing audio embeddings for the YouTube video and the Spotify song preview computed with Laion-CLAP [35]. The metadata includes an explicit flag to allow users to filter for explicit or non-explicit music.

2. **How many instances are there in total (of each type, if appropriate)?**

   - 15,296,232

3. **Does the dataset contain all possible instances or is it a sample (not necessarily random) of instances from a larger set?** If the dataset is a sample, then what is the larger set? Is the sample representative of the larger set (e.g., geographic coverage)? If so, please describe how this representativeness was validated/verified. If it is not representative of the larger set, please describe why not (e.g., to cover a more diverse range of instances, because instances were withheld or unavailable).

   - No, DISCO-10M does not cover all artists on Spotify and only a selection of popular songs of those that we do consider. The YouTube search results only contain 20 matches we take into consideration. To improve the dataset quality, we filter out matches that do not meet the threshold described in Section 3.2.

4. **What data does each instance consist of?** "Raw" data (e.g., unprocessed text or images) or features? In either case, please provide a description.

   - URLs to YouTube videos and Spotify song previews as well as song-specific metadata, such as artist names, artist/song IDs, YouTube video title/description snippet, video views, duration, Spotify song duration, and creation date

5. **Is there a label or target associated with each instance?** If so, please provide a description.

   - No.

6. **Is any information missing from individual instances?** If so, please provide a description explaining why this information is missing (e.g. because it was unavailable). This does not include intentionally removed information but might include, e.g., redacted text.

- The artist names are not always known since we do not have this information for every artist ID in our dataset. This is the case in 3.46% of all datapoints. In addition, we have 7.81% missing YouTube description snippets and 0.183% missing YouTube view counts.

7. **Are relationships between individual instances made explicit (e.g., users' movie ratings, social network links)?** If so, please describe how these relationships are made explicit.

   - No.

8. **Are there recommended data splits (e.g., training, development/validation, testing)?** If so, please provide a description of these splits, explaining the rationale behind them.

   - No.

9. **Are there any errors, sources of noise, or redundancies in the dataset?** If so, please provide a description.

   - We acknowledge the existence of duplicate songs stemming from different YouTube videos corresponding to the same Spotify song. These duplicates can be removed by filtering with stricter thresholds (cf. Section 3.2).

10. **Is the dataset self-contained, or does it link to or otherwise rely on external resources (e.g., websites, tweets, other datasets)?** If it links to or relies on external resources, a) are there guarantees that they will exist and remain constant over time; b) are there official archival versions of the complete dataset (i.e., including the external resources as they existed at the time the dataset was created); c) are there any restrictions (e.g., licenses, fees) associated with any of the external resources that might apply to a dataset consumer? Please provide descriptions of all external resources and any restrictions associated with them, as well as links or other access points, as appropriate.

    - DISCO-10M relies on the availability of the songs on YouTube and Spotify since we only link to those resources. The embeddings and other metadata are self-contained.

11. **Does the dataset contain data that might be considered confidential (e.g., data that is protected by legal privilege or by doctor– patient confidentiality, data that includes the content of individuals' non-public communications)?** If so, please provide a description.

    - No, there are no confidential datapoints in DISCO-10M.

12. **Does the dataset contain data that, if viewed directly, might be offensive, insulting, threatening, or might otherwise cause anxiety?** If so, please describe why.

    - DISCO-10M contains music with an explicit flag. We do not know in what ways the song is explicit (sexual, abusive, or others), but the flag allows users to filter for such songs easily. Additionally, DISCO-10M does not contain any links to age-restricted YouTube videos.

13. **Does the dataset identify any subpopulations (e.g., by age, gender)?** If so, please describe how these subpopulations are identified and provide a description of their respective distributions within the dataset.

    - No.

14. **Is it possible to identify individuals (i.e., one or more natural persons), either directly or indirectly (i.e., in combination with other data) from the dataset?** If so, please describe how.

    - Yes, the Spotify artist ID is directly related to one or multiple natural persons. Additionally, the YouTube video URLs we provide in the dataset are uploaded by one or multiple natural persons.

15. **Does the dataset contain data that might be considered sensitive in any way (e.g., data that reveals race or ethnic origins, sexual orientations, religious beliefs, political opinions or union member- ships, or locations; financial or health data; biometric or genetic data; forms of government identification, such as social security numbers; criminal history)?** If so, please provide a description.

    - No.

16. **Any other comments?**

- We emphasize the focus of our dataset on music and not on individuals. Additionally, we reiterate that this dataset is intended for research purposes only, as described in Section 5.

### J.3 Collection Process

1. **How was the data associated with each instance acquired?** Was the data directly observable (e.g., raw text, movie ratings), reported by subjects (e.g., survey responses), or indirectly inferred/derived from other data (e.g., part-of-speech tags, model-based guesses for age or language)? If the data was reported by subjects or indirectly inferred/derived from other data, was the data validated/verified? If so, please describe how.

   - The YouTube videos and metadata, and the Spotify tracks and metadata are observable and were collected by accessing the Spotify API as well as the YouTube API. The similarity scores and audio embeddings are computed by us.

2. **What mechanisms or procedures were used to collect the data (e.g., hardware apparatuses or sensors, manual human curation, software programs, software APIs)?** How were these mechanisms or procedures validated?

   - The Spotify API and the YouTube API. Our results were validated manually by assessing the quality of the retrieved information on random samples.

3. **If the dataset is a sample from a larger set, what was the sampling strategy (e.g., deterministic, probabilistic with specific sampling probabilities)?**

   - We started the Spotify artist scraping from the artist seed described in Appendix A. Additionally, we filter high-quality datapoints as described in Section 3.2.

4. **Who was involved in the data collection process (e.g., students, crowdworkers, contractors), and how were they compensated (e.g., how much were crowdworkers paid)?**

   - Only the authors of this paper were involved in the data collection process. *Author involvement and payment disclosed after acceptance.*

5. **Over what timeframe was the data collected?** Does this timeframe match the creation timeframe of the data associated with the instances (e.g., recent crawl of old news articles)? If not, please describe the time- frame in which the data associated with the instances was created.

   - January 2023 to June 2023 was the timeframe of data collection. The creation time of the songs is diverse and can be seen in Figure 4.

6. **Were any ethical review processes conducted (e.g., by an institutional review board)?** If so, please provide a description of these review processes, including the outcomes, as well as a link or other access point to any supporting documentation.

   - No.

7. **Did you collect the data from the individuals in question directly, or obtain it via third parties or other sources (e.g., websites)?**

   - We collected data from Spotify and YouTube, not from artists directly.

8. **Were the individuals in question notified about the data collection?** If so, please describe (or show with screenshots or other information) how notice was provided, and provide a link or other access point to, or otherwise reproduce, the exact language of the notification itself.

   - We did not notify any individuals about the data collection.

9. **Did the individuals in question consent to the collection and use of their data?** If so, please describe (or show with screenshots or other information) how consent was requested and provided, and provide a link or other access point to, or otherwise reproduce, the exact language to which the individuals consented.

   - We link to publicly available music on Spotify and YouTube. We allow every artist contained in our dataset to have their entries removed upon request.

10. **If consent was obtained, were the consenting individuals provided with a mechanism to revoke their consent in the future or for certain uses** If so, please provide a description, as well as a link or other access point to the mechanism (if appropriate).

- Artists have the possibility to search our dataset for their YouTube video links, and their Spotify artist IDs and track IDs. If artists wish to remove their content from YouTube or Spotify, they can contact those parties or remove it themselves; this would result in our links becoming invalid. Additionally, we allow artists to contact us at disco.dataset@ethz.ch to request the removal of their datapoints.

11. **Has an analysis of the potential impact of the dataset and its use on data subjects (e.g., a data protection impact analysis) been conducted?** If so, please provide a description of this analysis, including the outcomes, as well as a link or other access point to any supporting documentation.

    - Yes, we discuss the implications of our data collection pipeline and of our dataset in Appendix I.

12. **Any other comments?**

    - No.

### J.4 Preprocessing/cleaning/labeling

1. **Was any preprocessing/cleaning/labeling of the data done (e.g., discretization or bucketing, tokenization, part-of-speech tagging, SIFT feature extraction, removal of instances, processing of missing values)?**

    - We performed preprocessing by filtering, as described in Section 3.2. We do not process videos that are marked as age-restricted by YouTube, and we provide the explicit content flag from Spotify.

2. **Was the "raw" data saved in addition to the preprocessed/cleaned/labeled data (e.g., to support unanticipated future uses)?** If so, please provide a link or other access point to the "raw" data.

    - No.

3. **Is the software that was used to preprocess/clean/label the data available?** If so, please provide a link or other access point.

    - We use `spotipy` to access the Spotify API, `youtubesearchpython` to query the YouTube search, and `pytube` to access the video on YouTube.

4. **Any other comments?**

    - No.

### J.5 Uses

1. **Has the dataset been used for any tasks already?** If so, please provide a description.

    - No.

2. **Is there a repository that links to any or all papers or systems that use the dataset?** If so, please provide a link or other access point.

    - No.

3. **What (other) tasks could the dataset be used for?**

    - We encourage the research community to use the dataset for music analysis, video analysis, music information retrieval, generative models for music, music genre recognition, as well as other possible downstream tasks enabled by the provided embeddings.

4. **Is there anything about the composition of the dataset or the way it was collected and preprocessed/cleaned/labeled that might impact future uses?** For example, is there anything that a dataset consumer might need to know to avoid uses that could result in unfair treatment of individuals or groups (e.g., stereotyping, quality of service issues) or other risks or harms (e.g., legal risks, financial harms)? If so, please provide a description. Is there anything a dataset consumer could do to mitigate these risks or harms?

    - Yes, as discussed in Section 5.

5. **Are there tasks for which the dataset should not be used?** If so, please provide a description.

- We strongly advise to use DISCO-10M only for research purposes and not for commercial applications.

6. **Any other comments?**
    - No.

## J.6 Distribution

1. **Will the dataset be distributed to third parties outside of the entity (e.g., company, institution, organization) on behalf of which the dataset was created?** If so, please provide a description.
    - Yes, the dataset will be open-source.

2. **How will the dataset will be distributed (e.g., tarball on website, API, GitHub)?** Does the dataset have a digital object identifier (DOI)?
    - The dataset will be available on Hugging Face Datasets. DOI: 10.57967/hf/0754

3. **When will the dataset be distributed?**
    - Starting from 14.06.2023.

4. **Will the dataset be distributed under a copyright or other intellectual property (IP) license, and/or under applicable terms of use (ToU)?** If so, please describe this license and/or ToU, and provide a link or other access point to, or otherwise reproduce, any relevant licensing terms or ToU, as well as any fees associated with these restrictions.
    - CC-BY-4.0

5. **Have any third parties imposed IP-based or other restrictions on the data associated with the instances?** If so, please describe these restrictions, and provide a link or other access point to, or otherwise reproduce, any relevant licensing terms, as well as any fees associated with these restrictions.
    - We do not own the copyright of the music accessible through the provided links.

6. **Do any export controls or other regulatory restrictions apply to the dataset or to individual instances?** If so, please describe these restrictions, and provide a link or other access point to, or otherwise reproduce, any supporting documentation.
    - No.

7. **Any other comments?**
    - No.

## J.7 Maintenance

1. **Who will be supporting/hosting/maintaining the dataset?**
    - Hugging Face Datasets will host the dataset and we will maintain the dataset.

2. **How can the owner/curator/manager of the dataset be contacted (e.g., email address)?**
    - The authors can be contacted via disco.dataset@ethz.ch.

3. **Is there an erratum?** If so, please provide a link or other access point.
    - Not initially, will be started when necessary, and will be documented with future releases.

4. **Will the dataset be updated (e.g., to correct labeling errors, add new instances, delete instances)?** If so, please describe how often, by whom, and how updates will be communicated to dataset consumers (e.g., mailing list, GitHub)?
    - No, except for updates due to removal of dataset entries. Updates will be communicated on Hugging Face Datasets.

5. **If the dataset relates to people, are there applicable limits on the retention of the data associated with the instances (e.g., were the individuals in question told that their data would be retained for a fixed period of time and then deleted)?** If so, please describe these limits and explain how they will be enforced.

- Artists may contact us to have entries excluded from our dataset.

6. **Will older versions of the dataset continue to be supported/hosted/maintained?** If so, please describe how. If not, please describe how its obsolescence will be communicated to dataset consumers.

    - There are no older versions of DISCO-10M.

7. **If others want to extend/augment/build on/contribute to the dataset, is there a mechanism for them to do so?** If so, please provide a description. Will these contributions be validated/verified? If so, please describe how. If not, why not? Is there a process for communicating/distributing these contributions to dataset consumers? If so, please provide a description.

    - Updating and extending the dataset will be done on a case-by-case basis.

8. **Any other comments?**

    - No.

