# OpenReview forum: "DISCO-10M: A Large-Scale Music Dataset"
_NeurIPS.cc/2023/Track/Datasets_and_Benchmarks — NeurIPS 2023 Datasets and Benchmarks Poster_

### Official Review · Reviewer_pF6S · 2023-06-30

**Rating:** 5
**Confidence:** 3
**Clarity:** The paper is clearly written, but som…

**Strengths:**

The paper proposes a large-scale dataset which is expected to cover diversity of popular artists. The methods for forming the set of artists and songs and matching them with YouTube videos seem appropriately chosen. Overall this can be important contribution to the music information retrieval research community.


**Additional Feedback:**

In Section 3.2, it is not clear what "we observe an increase in density as the duration similarity decreases". Based on Figure 2.a the highest density value is when the similarity is close to 1.0, which does not match with the above statement.

Section 4 describes an evaluation of genre classification, but it is not clear how the quality is evaluated in this case.

**Correctness:**

It appears that the dataset is constructed in a sound way, but some verification of the obtained data would have made it more convincing. There is not much information about the results nor evaluation of the procedure that is used to find YouTube videos of each of the songs. The paper describes the empirically found procedure and parameters used to do the matching, but there is no objective information about how well this procedure works. How many of the songs were matched with a correct YouTube video? One would also expect that there is a bias in videos that are present in YouTube, so it would be useful to know if certain genres or artists are well present in the dataset or if some genres are missing. To understand how the quality of the matching procedure, it would be good to have information about what is contained in the Spotify preview and titles of Spotify songs vs. YouTube videos and how they affect the set of obtained matches. What kind of factors contribute to the differences in distances?


**Documentation:**

On a high level, there is sufficient information about the data collection. However, some important details are missing. The search of Spotify artists is initialized with a "hard-curated list of seed artists". How exactly has this list been obtained, and how well does it represent or affect the diversity of targeted music? How does the procedure that searches for related artists of each of the above affect or guarantee the diversity of artists? There is no information about what the Spotify related artist relationship represents, and how this affects the goals of the data collection.


**Ethics:**

The dataset depends on YouTube videos which formally cannot be used for research based on the Youtube license, but of course in reality many large-scale datasets are based on Youtube, so I don't see that as a big ethical issue.


**Limitations:**

Limitations regarding the data that depend on external resources (Spotify and YouTube) are well acknowledged. The limitations related to the data collection itself or analysis of the obtained data have not been discussed much.

**Opportunities For Improvement:**

The explanation and analysis of the data collection, which are the main points of the paper, is somewhat shallow. There does not seem to be any systematic quality control. The proposed methods  might be able to find a good sample of artists and YouTube videos of their popular songs, but currently, there is no verification about this. See review sections Correctness and Documentation for detailed comments.


**Relation To Prior Work:**

The relation to previous work is well described.

**Summary And Contributions:**

The manuscript proposes a new large-scale music dataset that is obtained by searching a large set of artists and their popular songs in Spotify, and then finding matching YouTube videos for songs based on duration, title, and audio similarity to Spotify previews.

---

> ### Author Response · Authors · 2023-08-14
>
> We thank the reviewer for their valuable feedback and address their concerns as follows:
>
> > How many of the songs were matched with a correct YouTube video?
>
> Accurately and objectively judging the match-quality of the dataset in a quantitative way proved difficult, as we have no underlying ground–truth labels for correct matches. The filters on the full dataset with ~15M entries were consciously chosen to be rather permissive to allow for further filtering based on the requirements for the downstream task at hand.
>
> > One would also expect that there is a bias in videos that are present in YouTube, so it would be useful to know if certain genres or artists are well present in the dataset or if some genres are missing.
>
> Figure 4 shows the genre distribution (according to their zero-shot classification) of songs in the dataset. Figure 7 (in the Appendix) extends this analysis to FMA genres. We found the classification with zero-shot classification to work best, as genres provided by Spotify are unreliable because they are supplied by the account holder and often vary widely in granularity or are simply missing. While we don’t expect the classification to be perfect, it should be sufficient to judge the overall distribution of genres in the dataset. The distribution of every music dataset is slightly different and we find it hard to make out a “representative” genre distribution. Should this be based on the number of artists per genre on Spotify? Or the number of monthly listeners per genre? Both change over time and introduce their own biases. We believe that the provided genre analysis makes researchers aware of the dataset distribution. If other distributions are desired, there is always the possibility to further filter the dataset with the available CLAP embeddings (for example by repeating our zero-shot classification and adjusting the choice of songs per genre or to filter for a specific genre).
>
> > To understand how the quality of the matching procedure, it would be good to have information about what is contained in the Spotify preview and titles of Spotify songs vs. YouTube videos and how they affect the set of obtained matches. What kind of factors contribute to the differences in distances?
>
> The [documentation](https://developer.spotify.com/documentation/web-api/reference/get-track) for obtaining the spotify preview url offers a brief explanation of what it is, a 30-second preview in MP3 format. Spotify does not provide more details on how this 30-second window is chosen.
>
> Factors contributing to the differences depend on the filtering step. For the duration similarity this is easily interpretable, while the others are based on neural embeddings that naturally allow for less interpretability. For the text embeddings this is based on BERT sentence embeddings, while the sound similarity is based on CLAP embeddings. For text embedding examples, we refer to Table 2 and Table 3, which showcase some values that we encountered.
>
> > The search of Spotify artists is initialized with a "hard-curated list of seed artists". How exactly has this list been obtained, and how well does it represent or affect the diversity of targeted music? How does the procedure that searches for related artists of each of the above affect or guarantee the diversity of artists?
>
> The curated list of seed artists was determined heuristically based on popular artists for common genres from GTZAN. We expect that the initial selection of seed artists will have less impact as we expanded our search sufficiently to include more and more representative and diverse artists. The [documentation](https://developer.spotify.com/documentation/web-api/reference/get-an-artists-related-artists) states that the related artist graph is created from similarity based on analysis of the Spotify community's listening history. In summary, we are aware that our approach is heavily dependant on Spotify and its provided features, such as the relationship graph of artists, as well as the data provided by YouTube and its search algorithm. However, we believe that relying on the largest and most popular platforms leads to a representative cross-section of what content is currently consumed by users.
>
> > In Section 3.2, it is not clear what "we observe an increase in density as the duration similarity decreases". Based on Figure 2.a the highest density value is when the similarity is close to 1.0, which does not match with the above statement.
>
> We wanted to refer to the small peak at approximately 0.05 in Figure 2 a). We clarified this in the revised manuscript.
>
> > Section 4 describes an evaluation of genre classification, but it is not clear how the quality is evaluated in this case.
>
> As mentioned above, we do not have ground-truth data to analyze the quality of the predictions. However, we show consistent classification between GTZAN and FMA genres (plots can be found in the Appendix).

---

> > ### Comment · Reviewer_pF6S · 2023-08-29
> >
> > Thank you for the explanations. Because the quality of the matches cannot be verified in any way, I will keep my original recommendation.

---

### Official Review · Reviewer_LrKY · 2023-07-21
**A new music dataset for scalable research**

**Rating:** 7
**Confidence:** 5
**Correctness:** Yes
**Clarity:** This paper is clearly written.

**Strengths:**

1. This paper's primary and significant contribution lies in providing a new, extensive music dataset. The domain of music research has long struggled with a scarcity of publicly available large-scale data, hindering progress and limiting the scope of investigations. By presenting DISCO-10M, the paper empowers researchers with a valuable resource to explore scalable music informatics research.

2. Another strength of this work is its clarity. The paper is well-motivated and easy to follow. All the data creation processes are clearly described, which enables reproducible research. Also, the paper explicitly addresses the limitations and potential ethical issues of the proposed dataset.

**Additional Feedback:**

- Table 1 already provides a lot of information. But more attributes, such as licenses and available labels (e.g., tags) can be helpful.
- Table 1. MagnaTagATune dataset (~26k) has also been broadly used as scalable music data. Also, the Music4All dataset is relevant.
- If I understood correctly, the number of unique tracks is 2.5M. I believe this needs to be more clearly addressed.
- The authors used CLAP embeddings to measure the acoustic similarity. I'm curious if it's a better choice than using fingerprint algorithms.


**Documentation:**

Yes

**Ethics:**

There are possible license issues with using the proposed data. However, the authors only distributed URLs of publicly available content. The dataset itself is fine, but the usage might include potential issues.

**Limitations:**

Yes

**Opportunities For Improvement:**

1. While Section 4 provides a description of the data distribution, I believe a more comprehensive comparison of DISCO-10M with other existing datasets will further strengthen its impact and significance. By delving into differences in genre distribution, released years, and overlapping tracks, the paper can establish a clearer picture of the dataset's unique attributes.

2. Furthermore, the authors can choose one task (self-supervised learning, semi-supervised learning, or generative models) to demonstrate the potential of DISCO-10M and its usability.

**Relation To Prior Work:**

Relation to prior work is clearly discussed.

**Summary And Contributions:**

The authors propose a new large-scale music dataset, DISCO-10M, which comprises 11M audio clips from 400k artists. The paper depicts the details of data collection and filtering, then discusses data distribution and limitations. The proposed dataset can be used in a broad range of music informatics research, including music analysis, recommender systems, and generative models.

Contribution
- Large-scale music data with up-to-date tracks
- Detailed guide for data collection and filtering

---

> ### Author Response · Authors · 2023-08-14
>
> We thank the reviewer for their valuable feedback and address their concerns as follows:
>
> > 1. While Section 4 provides a description of the data distribution, I believe a more comprehensive comparison of DISCO-10M with other existing datasets will further strengthen its impact and significance. By delving into differences in genre distribution, released years, and overlapping tracks, the paper can establish a clearer picture of the dataset's unique attributes.
>
> In addition to the description of the dataset distribution in Section 4, we provide a comparative high-level overview of the datasets in Table 1. We further extended this comparison in Section 2 with the MagnaTagATune and Music4All datasets in the revised manuscript. Due to the different nature of the datasets and the features they provide we find it hard to compare them in a unified way. Some datasets do not provide genres for songs or release years and overlapping tracks are not applicable to all of them.
>
> > 2. Furthermore, the authors can choose one task (self-supervised learning, semi-supervised learning, or generative models) to demonstrate the potential of DISCO-10M and its usability.
>
> We agree that a demonstration of DISCO-10M’s capabilities would be a valuable addition. Although we show the usefulness of zero-shot classification based on the embeddings for the genre classification, further research is required to train large models on DISCO-10M to make use of its unique scale. We strongly believe that releasing DISCO-10M to the research community is of great value and enables the community to train models based on available data, in contrast to closed-source datasets that only few parties have access to.
>
> > Table 1 already provides a lot of information. But more attributes, such as licenses and available labels (e.g., tags) can be helpful.
>
> Table 1 gives a good overview of existing datasets. More information for every dataset is discussed in Section 2. In our opinion, adding more information to Table 1 would hurt readability and comprehensiveness.
>
> > Table 1. MagnaTagATune dataset (~26k) has also been broadly used as scalable music data. Also, the Music4All dataset is relevant.
>
> Thank you for pointing out these relevant prior works. We have added both MagnaTagATune and Music4All to the related work section of the revised manuscript.
>
> > If I understood correctly, the number of unique tracks is 2.5M. I believe this needs to be more clearly addressed.
>
> We clarified this fact further in the revised manuscript.
>
> > The authors used CLAP embeddings to measure the acoustic similarity. I'm curious if it's a better choice than using fingerprint algorithms.
>
> CLAP embeddings are used in one of four filtering steps (the other being based on text and duration) and are not very restrictive. Furthermore, as we provide the embeddings alongside the dataset, the same filtering can be applied with a more restrictive threshold when needed.

---

### Official Review · Reviewer_fHP3 · 2023-07-22
**Large Dataset for Music Technology**

**Rating:** 5
**Confidence:** 5
**Correctness:** I believe the descriptions in the pap…
**Clarity:** well written and easy to follow

**Strengths:**

- the number of songs in the dataset is unprecedented scale
- the curation scheme sounds straightforward and easy to reproduce
- CLAP embeddings of songs in the dataset would save time and computes of people who use this dataset
- similar songs in the dataset are taken care of by the filtering scheme, which ensures the diversity and quantity of dataset

**Additional Feedback:**

As noted in the weakness section, I strongly believe this paper needs some evidence on whether the dataset is really helpful for music tasks. At the current state, readers have no idea if using the dataset can really lead to an improvement in some task or not

**Documentation:**

I believe users can reproduce the dataset

**Ethics:**

A copyright issue is not addressed well in the paper. In most countries, it is not allowed to copy and use copyrighted songs for training. The machine learning community should take it seriously and consider how using the copyrighted data is beneficial to artists. The problem I see in the paper is the above concern is highlighted very well

**Limitations:**

- the author properly addressed the issue that the links might be expired in future, which highlight the drawback of not having audio signals in the dataset. However, I believe this will not impact the total amount for some years to come


**Opportunities For Improvement:**

- any evaluation is provided to prove releasing the dataset is beneficial to the community
- CLAP is trained with contrastive learning and suboptimal to similarity search. The author should justify why CLAP is selected in the paper
- the audio quality of songs in the dataset is not clarified well (e.g., fs, bitrate, bit depth, channels, etc.)


**Relation To Prior Work:**

prior works are covered very nicely

**Summary And Contributions:**

The paper introduces unprecedented amount of music data for training large-scale music models. The dataset is made of careful filtering of curated music from Spotify and YouTube music, which includes similarity filtering in the CLAP embedding space. Data analysis is conducted to see how different genres are mapped uniformly in the embedding space and to prove how diverse the dataset is.

---

> ### Author Response · Authors · 2023-08-14
>
> We thank the reviewer for their valuable feedback and address their concerns as follows:
>
> > any evaluation is provided to prove releasing the dataset is beneficial to the community
>
> We present a new dataset at an unprecedented scale with additional sound embeddings for improved accessibility. Especially for generative models, it is widely acknowledged that the scale and quality of the dataset are crucial. DISCO-10M, therefore, is a prime candidate for these tasks. However, we currently lack the necessary computational resources and time to train such large generative models.
>
> Many of the recent advances in music synthesis were led by organizations with large closed-source datasets. We believe releasing DISCO-10M to the community will democratize future research by making a large-scale dataset freely available.
>
> Furthermore, to bridge the gap between academia and industry for MIR tasks, it is important that researchers have access to large-scale datasets on which to test their algorithms, and since it has been over a decade since the last order of magnitude increase in dataset size (Million Song Dataset) we believe DISCO-10M is a logical next step.
>
> > CLAP is trained with contrastive learning and suboptimal to similarity search. The author should justify why CLAP is selected in the paper
>
> We do not do similarity search in the paper but rather similarity-based filtering on pairs of audio snippets. The filtering, based on CLAP embeddings, is one of four and mainly used to filter out bad matches. Users of the dataset are free to further filter the ~15 million entries based on other strategies. We decided to use CLAP as it has the same properties as CLIP, but adapted to the audio setting. Similarity comparison is a common use case for CLIP embeddings which also transfers to CLAP. Another reason to use CLAP for similarity filtering is that we already provide these embeddings with the dataset.
>
> > the audio quality of songs in the dataset is not clarified well (e.g., fs, bitrate, bit depth, channels, etc.)
>
> We provide the audio specification in the following table, where the distribution of different settings is given in brackets. We have added this information in the revised manuscript.
>
> |  audio source |  bits per sample |  sample rate | channels  |  bitrate |
> |---|---|:---|:---|---:|
> |  YouTube |  16  |  44.1k (99%), 48k (0.04%)  | 1(0.18%), 2(99%), 6(0.04%)  |  - |
> |  Spotify |  16  |  44.1k |  2 |  96k |
>
>
> > A copyright issue is not addressed well in the paper. In most countries, it is not allowed to copy and use copyrighted songs for training. The machine learning community should take it seriously and consider how using the copyrighted data is beneficial to artists. The problem I see in the paper is the above concern is highlighted very well
>
> We are aware of the issue and, therefore, only provide the dataset for scientific research and not for commercial use.

---

> > ### Comment · Reviewer_fHP3 · 2023-08-29
> >
> > Thanks for your reply.
> >
> > regarding not having any evaluation in the paper, I guess you could have tried some downstream tasks as you computed CLAP embeddings?
> >
> > About the copyright issue, I disagree that it is beneficial for the ML community to simply get access to large data (owned by someone else) without seeing any evidence that the data can actually enhance many tasks. From the ethical prospect, in the current paper, it sounds to song creators like someone comes to their farms, steal their vegetables without proving the benefit why the stolen vegetables help many people. You should prove its effectiveness while saying something like these vegetables will save a lot of people who have disease?

---

> > > ### Author Response · Authors · 2023-08-30
> > >
> > > We thank the reviewer for their comment. Regarding downstream tasks, we agree that more applications would be useful. However, we already show how music genre recognition and text embedding searches can be useful tools to navigate the dataset and gain further insights. Furthermore, members of the community have already begun using the dataset for downstream tasks, such as music-style blending [1].
> > >
> > > We understand the reviewer's concern regarding copyright and want to highlight our strong focus on non-commercial research. In the realm of non-commercial research, it is important to acknowledge the legal framework that supports our approach. Various legal statements issued by governmental institutions underscore the significance of fair use in such contexts [2, 3]. Fair use, a well-established legal doctrine, provides a solid foundation for using copyrighted material in non-commercial research. This doctrine is recognized as an important exception that enables researchers to access and analyze copyrighted material without seeking explicit permission from the copyright holders. In this regard we follow in the footsteps of previously released datasets which also abide by copyright laws [4, 5, 6].
> > >
> > >
> > > [1] https://github.com/andrewgcodes/blend
> > >
> > > [2] https://www.copyright.gov/fair-use/
> > >
> > > [3] https://eur-lex.europa.eu/eli/dir/2019/790/oj
> > >
> > > [4] https://laion.ai/faq/
> > >
> > > [5] http://research.google.com/audioset/download.html
> > >
> > > [6] http://millionsongdataset.com/

---

### Official Review · Reviewer_28sH · 2023-07-22
**Good paper**

**Rating:** 7
**Confidence:** 5
**Correctness:** The claims are generally correct.
**Clarity:** Clearly-written paper.

**Strengths:**

The paper designs a pipeline to jointly use the information from Spotify and Youtube, and proposes several similarity measures to filter the audios.

**Additional Feedback:**

NA

**Documentation:**

Enough documentaion.

**Ethics:**

Ethical issues mainly include copyright, which has been discussed in the attachment.

**Limitations:**

1. Other important meta information such as the genre and lyrics is missing, and such information can be also important for music information retrieval and music generation.

2. The total duration of the dataset is unknown.





**Opportunities For Improvement:**

See below.

**Relation To Prior Work:**

Discsussions are adequate.

**Summary And Contributions:**

This paper develops a large scale music dataset by collecting the large amount of musical audio data from Youtube and Spotify. The resulting dataset consists of over 11 million audio clips. Pipelines of getting the music list and pilcking out the candidate audios are carefully designed. CLAP embeddings are also computed.

---

> ### Author Response · Authors · 2023-08-14
>
> We thank the reviewer for their feedback. We address the mentioned limitations as follows:
>
> > 1. Other important meta information such as the genre and lyrics is missing, and such information can be also important for music information retrieval and music generation.
>
> We agree that lyrics could be an interesting additional feature of the dataset. However, we cannot provide lyrics directly due to copyright laws. Regarding genres, we found the genres provided by Spotify to be highly inconsistent as they are chosen by the artists themselves and can vary greatly in consistency and granularity. Therefore, they are not part of our dataset. Nevertheless, the paper demonstrates that zero-shot genre classification using the CLAP embeddings provided with the dataset is viable for inferring genres.
>
> > 2. The total duration of the dataset is unknown.
>
> The total duration of the YouTube music tracks linked in the dataset is approximately 1,062,604 hours, which is more than 121 years. We have added this information in the revised manuscript.

---

### Author Response · Authors · 2023-08-28

We thank the reviewers for their time and effort spent on reviewing our work. The feedback we received has been very valuable, and we have incorporated it into the revised version, which was completed two weeks ago. The main changes included updating the related work, making various readability improvements, and adding extra information to the appendix. We believe that these changes helped to strengthen the paper, and we are thankful for the positive points highlighted.

---

### Decision · Program_Chairs · 2023-09-22

**Decision:**

Accept (Poster)

**Comment:**

This paper presents DISCO-10M, a large-scale music dataset to help advance the development of novel machine learning models for music. The dataset was collected through a careful filtering pipeline of curated music from Spotify and YouTube music. The authors also provide CLAP embeddings for DISCO-10M, allowing efficient exploration of machine learning applications. The reviewers agree about the usefulness of the dataset, but raise a concern about copyright issue of music. The authors highlight that they do not own the copyright of any music referenced in the dataset, thus only provide url links to the publicly available sources. In a positive view, this allows dataset users themseleves to assess the adequateness of DISCO-10M for research. The paper also provides a reference pipeline for them to collect other copyright-free music from the Internet.